# On the joint role of non-Hispanic Black race/ethnicity and weight status in predicting postmenopausal weight gain

Christopher N. Ford[1]*, Shine Chang[2], Alexis C. Wood[3], JoAnn E. Manson[4], David O. Garcia[5], Helena Laroche[6], Chloe E. Bird[7], Mara Z. Vitolins[8]

1 Rush Institute for Healthy Aging, Rush University Medical Center, Chicago, IL, United States of America, 2 Department of Epidemiology, University of Texas MD Anderson Cancer Center, Houston, TX, United States of America, 3 USDA/ARS Children's Nutrition Research Center, Baylor College of Medicine, Houston, TX, United States of America, 4 Department of Medicine, Brigham and Women's Hospital, Harvard Medical School, Boston, MA, United States of America, 5 Department of Health Promotion Sciences, Mel and Enid Zuckerman College of Public Health, University of Arizona, Tucson, AZ, United States of America, 6 Center for Children's Healthy Lifestyles and Nutrition, Children's Mercy Hospital, Kansas City and University of Missouri, Kansas City, MO, United States of America, 7 RAND Corporation, Santa Monica, CA, United States of America, 8 Department of Epidemiology and Prevention, Wake Forest School of Medicine, Wake Forest, NC, United States of America

* Christopher_Ford@Rush.edu

## Abstract

### Objectives

To determine how baseline weight status contributes to differences in postmenopausal weight gain among non-Hispanic Blacks (NHBs) and non-Hispanic Whites (NHWs).

### Methods

Data were included from 70,750 NHW and NHB postmenopausal women from the Women's Health Initiative Observational Study (WHI OS). Body Mass Index (BMI) at baseline was used to classify women as having normal weight, overweight, obese class I, obese class II or obese class III. Cox proportional hazards was used to estimate the hazard of a 10% or more increase in weight from baseline.

### Results

In both crude and adjusted models, NHBs were more likely to experience ≥10% weight gain than NHWs within the same category of baseline weight status. Moreover, NHBs who were normal weight at baseline were most likely to experience ≥10% weight gain in both crude and adjusted models. Age-stratified results were consistent with overall findings. In all age categories, NHBs who were normal weight at baseline were most likely to experience ≥10% weight gain. Based on the results of adjusted models, the joint influence of NHB race/ethnicity and weight status on risk of postmenopausal weight gain was both sub-additive and sub-multiplicative.

**Data Availability Statement:** The data, which is third-party owned by the Women's Health Initiative (WHI), can be accessed through the WHI website

(https://www.whi.org/page/propose-a-paper) with
an approved manuscript proposal. The authors had
no special access privileges to the data.

**Funding:** The Women's Health Initiative program is
funded by the National Heart, Lung, and Blood
Institute, National Institutes of Health, U.S.
Department of Health and Human Services through
contracts HHSN268201100046C,
HHSN26801100001C, HHSN268201100002C,
HHSN268201100003C, HHSN268201100004C,
and HHSN271201100004C. CF and SC received
salary support from the National Institutes of
Health, National Cancer Institute (5 R25
CA057730-24). CB received salary support from
the RAND Corporation. The funders had no role in
the study design, data collection and analysis,
decision to publish, or preparation of the
manuscript. The specific roles of these authors are
articulated in the 'author contributions' section.

**Competing interests:** CEB is employed by the
RAND Corporation. This does not alter our
adherence to PLOS ONE policies on sharing data
and materials.

## Conclusion

NHBs are more likely to experience postmenopausal weight gain than NHWs, and the disparity in risk is most pronounced among those who are normal weight at baseline. To address the disparity in postmenopausal obesity, future studies should focus on identifying and modifying factors that promote weight gain among normal weight NHBs.

## Introduction

Non-Hispanic Blacks (NHB) in the US have higher rates of obesity than non-Hispanic Whites (NHW) [1], which is thought to underlie disparities in chronic disease risk [2–4]. Individuals with obesity are at increased risk of cardiometabolic diseases including coronary heart disease [5], stroke [6], and diabetes [7, 8]. In women ages 60 and older, a significantly higher proportion of NHBs have obesity by body mass index (BMI) than NHWs. The most recently available estimates from the National Health and Nutrition Examination Survey show that 57.5% percent of NHBs have obesity, compared with 38.2% of NHWs [9]. Previous studies reported greater risk of weight gain in NHBs compared to NHWs at earlier life stages. In young-, middle-, and older- aged women, NHBs were at greater risk of weight gain than NHWs [10–13], but few studies have examined whether there are differences in the risk of postmenopausal weight gain in NHBs and NHWs. Weight status influences risk of weight gain and significantly higher rates of overweight, obesity and extreme obesity in NHBs compared to NHWs age 60 and older have been previously noted [14]. However, it is unclear how race/ethnicity and weight status jointly influence risk of postmenopausal weight gain in NHBs and NHWs.

   This study uses data from the Women's Health Initiative Observational Study (WHI OS) to determine whether there are differences between NHBs and NHWs in the risk of postmenopausal weight gain, and characterize the role of baseline weight status. We compare the risk of ≥10% weight gain in NHBs and NHWs overall, examine the interaction of race/ethnicity and baseline weight status, and determine the extent to which differences in risk are explained by differences in baseline weight status with and without adjustment for potential confounders.

## Methods

### Data and sample

Data were used from the WHI OS, which consists of 93,676 postmenopausal women who were enrolled between September 1993 and December 1998 and followed for up to eight years [15]. The analytic sample was restricted to NHW and NHB women (n = 85,651) in order to specifically examine how NHB race/ethnicity and baseline weight status jointly influence risk of postmenopausal weight gain. We excluded those with underweight BMI (<18.5 kg/m2) at baseline (n = 955) in light of the potential for confounding of the relationship between race/ethnicity and weight gain by chronic diseases associated with wasting [16]. We also excluded those who self-identified as diabetic at baseline (n = 3,291), and those who reported a history of cancer (n = 10,655) thereby resulting in a final analytic sample comprising 70,750 respondents from the WHI OS. Written informed consent was obtained from all respondents. All study procedures were approved by institutional review boards at each of 40 participating clinical centers. A complete list of participating clinical centers is available elsewhere [17]. All data were deidentified before the authors had access to it. This secondary analysis was approved by the Institutional Review Board of Rush University Medical Center.

WHI OS data also included sociodemographic information that was collected at baseline using a standard questionnaire. This information included annual household income (less than $10,000; $10,000 to $19,999; $20,000 to $34,999; $35,000 to $49,999; $50,000 to $74,999; $75,000 to $99,999; $100,000 to $149,999; and $150,000 or more), race/ethnicity (American Indian or Alaskan Native, Asian or Pacific Islander, Black or African American, Hispanic/ Latino, White [not of Hispanic Origin], or Other), and age (computed from birth date ascertained at screening). Usual alcohol intake was assessed at baseline using a standardized questionnaire. Possible responses ranged from 'non-drinker' to '7 or more drinks per week' (non-drinker, past drinker, <1 drink per month, <1 drink per week, 1–6 drinks per week, or 7 or more drinks per week). Time spent engaging in mild, moderate, and vigorous intensity physical activity was assessed at baseline using a questionnaire which has been described in detail elsewhere [18]. Mild physical activity was defined as walking, while moderate intensity activity was defined as 'not exhausting' and included biking outdoors, using a stationary exercise machine, calisthenics, easy swimming and dancing. Strenuous or hard exercise was defined as activities in which 'You work up a sweat and your heart beats fast' and included activities like aerobics, aerobic dancing, jogging, tennis, and swimming laps. Usual dietary intake during the past year was assessed at baseline and year three of follow-up using a Food Frequency Questionnaire comprising 122 items.

## Approach

BMI at baseline, computed using measured height and weight, was used to classify respondents as normal weight (BMI: 18.5–24.9 kg/m$^2$), overweight (BMI: 25.0–29.9 kg/m$^2$), obese class I (BMI: 30.0–34.9 kg/m$^2$), obese class II (BMI: 35.0–39.9 kg/m$^2$), or obese class III (BMI $\geq$40.0 kg/m$^2$). Weight was subsequently measured only at year one of follow-up, while self-reported 'highest weight since last follow-up' was available at follow-up years one through eight. Measured weight and reported 'highest weight since last follow-up at follow-up year one were found to be highly correlated (Pearson's $r$: 0.97; $p$<0.001). Thus, self-reported 'highest weight since last follow-up' was used to characterize the outcome variable, defined as $\geq$10% increase in weight from baseline weight.

Multiple imputation with chained equations was used to impute missing values in 10 data sets. The choice to impute 10 datasets was made as a conservative application of the approach used by Gao, Wilson and Hepgul et al., who reported imputing 20 copies of their data in their 2020 study published in the Journal of the American Medical Association [3]. In sensitivity analyses, we compared our primary results with and without imputation (see S3 Table).

[19]. Analyses were repeated in each data set and estimates were pooled [20]. Following previous works using data from the WHI [21–23], Cox proportional hazards models were used to estimate the relationship between race/ethnicity and a $\geq$10.0% weight gain from baseline. Respondents' self-reported highest weight since last follow-up was measured at one, three, four, five, six, seven and eight years of follow-up. The proportional hazards assumption was tested using plots of log-log(survival) vs. log(follow-up time), and the proportional hazards assumption was deemed to be satisfied if group-specific plots (e.g., race/ethnicity) were approximately parallel [24]. Overall hazard ratios comparing NHBs to NHWs, and comparing successive categories of weight status to those who were normal weight at baseline, were computed.

To evaluate the interaction of race/ethnicity and weight status on the additive and multiplicative scales, hazard ratios were computed comparing each combination of race/ethnicity and baseline weight status to a common referent group (normal weight NHWs) using an appropriate categorical interaction term. Following Hosmer and Lemeshow (1992), departure from

additive interaction between NHB race/ethnicity and weight status was evaluated using the general formula:

$$(H_{1k}/H_{00}) - (H_{10}/H_{00}) - (H_{0k}/H_{00}) + 1,$$

where *k* represents category of baseline weight status [25]. Departure from multiplicative interaction was assessed using the following formula:

$$(H_{1k}/H_{00})/[(H_{10}/H_{00}) * (H_{0k}/H_{00}).$$

If the interaction between race/ethnicity and weight status is additive, then joint impact of these variables on risk of postmenopausal weight gain is equal to the sum of their individual impact. However, if the relationship is multiplicative, then race/ethnicity and weight status jointly influence the risk of postmenopausal weight gain and together have greater influence than the sum of each variable's independent contribution to the overall risk relationship.

### Interaction and confounding

Interaction between race/ethnicity (comprising NHWs and NHBs) and weight status at baseline was examined by including an appropriate interaction term and evaluating the resulting Wald test p-value associated with this coefficient. The interaction of race/ethnicity and baseline BMI was significant (p<0.001). Backward selection (α = 0.10) was used to identify potential confounders to be included in adjusted models. The fully-saturated model included education level, annual household income, smoking status, alcohol intake, age, total energy intake at baseline and MET-hours of mild, moderate and hard exercise. An equivalent set of potential confounders was obtained using forward selection (α = 0.10) when variables were introduced in the opposite order used in the backward selection model. In both approaches, the models were constrained to include a term denoting the interaction of race/ethnicity and baseline weight status. Final adjusted models controlled for education level, annual household income, smoking status, alcohol intake, age, total energy intake at baseline and MET-hours of mild, moderate and hard exercise. All analyses were carried out in Stata (Version 16, Stata Corp, College Station, Texas, USA).

## Results

Selected sample characteristics are given in Table 1. The majority of respondents were NHW (91.4%), and normal weight (40.7%) or overweight (34.2%) at baseline, with a mean age of 63.5 years (± 7.3 years). On average, NHB respondents were younger, had greater rates of class I, II, and III obesity at baseline, and were more likely to have an annual household income of less than $20,000 (27.4%) than NHWs (*p*<0.001). NHBs were also more than twice as likely as NHWs to report non-drinking status (*p*<0.001).

Log-log(survival) was plotted against log(follow-up time) by race/ethnicity to determine whether the proportional hazards assumption was met. The plots were approximately parallel, thereby confirming that the proportionality assumption was met.

Unadjusted smoothed hazards by race/ethnicity are given in Fig 1. As shown, NHBs were 1.54 times (95% CI: 1.46, 1.62) more likely to experience ≥10% weight gain than NHWs.

Unadjusted smoothed hazards by baseline weight status are presented in S1 Fig. As shown, women with class I obesity (HR: 1.05; 95% CI: 1.00, 1.10) and class II obesity (HR: 1.17; 95% CI: 1.09, 1.26) at baseline were more likely to experience ≥10% weight gain than those with normal weight at baseline.

Table 2 shows overall hazard ratios comparing risk of weight gain by weight status to normal weight respondents. In unadjusted overall models, those with class II obesity at baseline

**Table 1. Selected characteristics of non-Hispanic White and non-Hispanic Black respondents from the Women's Health Initiative Observational Study overall, and by race/ethnicity[1].**

| | Overall | Non-Hispanic White | Non-Hispanic Black | p-value |
|---|---|---|---|---|
| | <-------------------------------------------N (%)-------------------------------------------> | | | |
| N | 70,750 | 64,676 (91.4%) | 6,074 (8.6%) | |
| **Weight status** | | | | |
| Normal weight (BMI: 18.5–24.9 kg/m$^2$) | 28,773 (40.7%) | 27,597 (42.7%) | 1,176 (19.4%) | <0.001 |
| Overweight (BMI: 25.0–29.9 kg/m$^2$) | 24,187 (34.2%) | 22,110 (34.2%) | 2,077 (34.2%) | |
| Obese class I (BMI: 30.0–34.9 kg/m$^2$) | 10,770 (15.2%) | 9,270 (14.3%) | 1,500 (24.7%) | |
| Obese class II (BMI: 35.0–39.9 kg/m$^2$) | 3,867 (5.5%) | 3,178 (4.9%) | 689 (11.3%) | |
| Obese class III (BMI $\geq$40.0 kg/m$^2$) | 2,318 (3.3%) | 1,771 (2.7%) | 547 (9.0%) | |
| Missing | 835 (1.2%) | 750 (1.2%) | 85 (1.4%) | |
| **Waist circumference** | | | | |
| <88 cm | 47,012 (66.4%) | 44,022 (68.1%) | 2,990 (49.2%) | <0.001 |
| $\geq$88 cm | 23,431 (33.1%) | 20,359 (31.5%) | 3,072 (50.6%) | |
| Missing | 307 (0.4%) | 295 (0.5%) | 12 (0.2%) | |
| **Highest education completed** | | | | |
| Less than high school | 14,181 (20.0%) | 12,599 (19.5%) | 1,582 (26.0%) | <0.001 |
| High school diploma or equivalent | 25,566 (36.1%) | 23,353 (36.1%) | 2,213 (36.4%) | |
| Some college | 8,301 (11.7%) | 7,772 (12.0%) | 529 (8.7%) | |
| Baccalaureate degree or more | 22,142 (31.3%) | 20,472 (31.7%) | 1,670 (27.5%) | |
| Missing | 560 (0.8%) | 480 (0.7%) | 80 (1.3%) | |
| **Household income** | | | | |
| Less than $20,000 | 9,390 (13.3%) | 7,724 (11.9%) | 1,666 (27.4%) | <0.001 |
| $20,000 to $49,999 | 28,554 (40.4%) | 26,273 (40.6%) | 2,281 (37.6%) | |
| $50,000 to $99,999 | 20,153 (28.5%) | 18,839 (29.1%) | 1,314 (21.6%) | |
| $100,000 or more | 7,608 (10.8%) | 7,334 (11.3%) | 274 (4.5%) | |
| Missing | 5,045 (7.1%) | 4,506 (7.0%) | 539 (8.9%) | |
| **Smoking status** | | | | |
| Never | 34,793 (49.2%) | 31,782 (49.1%) | 3,011 (49.6%) | <0.001 |
| Former | 30,644 (43.3%) | 28,364 (43.9%) | 2,280 (37.5%) | |
| Current | 4,323 (6.1%) | 3,678 (5.7%) | 645 (10.6%) | |
| Missing | 990 (1.4%) | 852 (1.3%) | 138 (2.3%) | |
| **Alcohol use** | | | | |
| Non-drinker | 6,713 (9.5%) | 5,610 (8.7%) | 1,103 (18.2%) | <0.001 |
| Past drinker | 12,183 (17.2%) | 10,277 (15.9%) | 1,906 (31.4%) | |
| <1 drink per month | 8,063 (11.4%) | 7,289 (11.3%) | 774 (12.7%) | |
| <1 drink per week | 14,518 (20.5%) | 13,452 (20.8%) | 1,066 (17.6%) | |
| 1 to 6 drinks per week | 19,236 (27.2%) | 18,393 (28.4%) | 843 (13.9%) | |
| 7 or more drinks per week | 9,560 (13.5%) | 9,280 (14.3%) | 280 (4.6%) | |
| Missing | 477 (0.7%) | 375 (0.6%) | 102 (1.7%) | |
| **Age category** | | | | |
| 49–54 years | 9,458 (13.4%) | 8,335 (12.9%) | 1,123 (18.5%) | <0.001 |
| 55–59 years | 13,402 (18.9%) | 12,066 (18.7%) | 1,336 (22.0%) | |
| 60–64 years | 15,633 (22.1%) | 14,140 (21.9%) | 1,493 (24.6%) | |
| 65 and older | 32,257 (45.6%) | 30,135 (46.6%) | 2,122 (34.9%) | |
| Missing | 0 (0.0%) | 0 (0.0%) | 0 (0.0%) | |
| | <----------------------------Mean ± standard deviation----------------------------> | | | |

*(Continued)*

**Table 1.** (Continued)

|  | Overall | Non-Hispanic White | Non-Hispanic Black | p-value |
|---|---|---|---|---|
| Age | 63.5 ± 7.3 | 63.6 ± 7.3 | 61.8 ± 7.3 | <0.001 |
| Missing | 0 (0.0%) | 0 (0.0%) | 0 (0.0%) |  |
| Physical activity (MET-hours/week) |  |  |  |  |
| Mild exercise | 1.4 ± 3.1 | 1.4 ± 3.2 | 0.7 ± 2.2 | <0.001 |
| Missing | 804 (1.1%) | 736 (1.1%) | 68 (1.1%) |  |
| Moderate exercise | 3.4 ± 5.4 | 3.5 ± 5.5 | 2.3 ± 4.5 | <0.001 |
| Missing | 804 (1.1%) | 736 (1.1%) | 68 (1.1%) |  |
| Hard exercise | 4.0 ± 8.5 | 4.0 ± 8.6 | 3.4 ± 7.8 | <0.001 |
| Missing | 804 (1.1%) | 736 (1.1%) | 68 (1.1%) |  |
| Total daily energy intake | 1,555 ± 666 | 1,559 ± 619 | 1,503 ± 1,045 | <0.001 |
| Missing | 64 (0.1%) | 56 (0.1%) | 8 (0.1%) |  |

[1] P-values given correspond to a $X^2$ test for categorical variables, and to a Students t-test for continuous variables (age and physical activity).

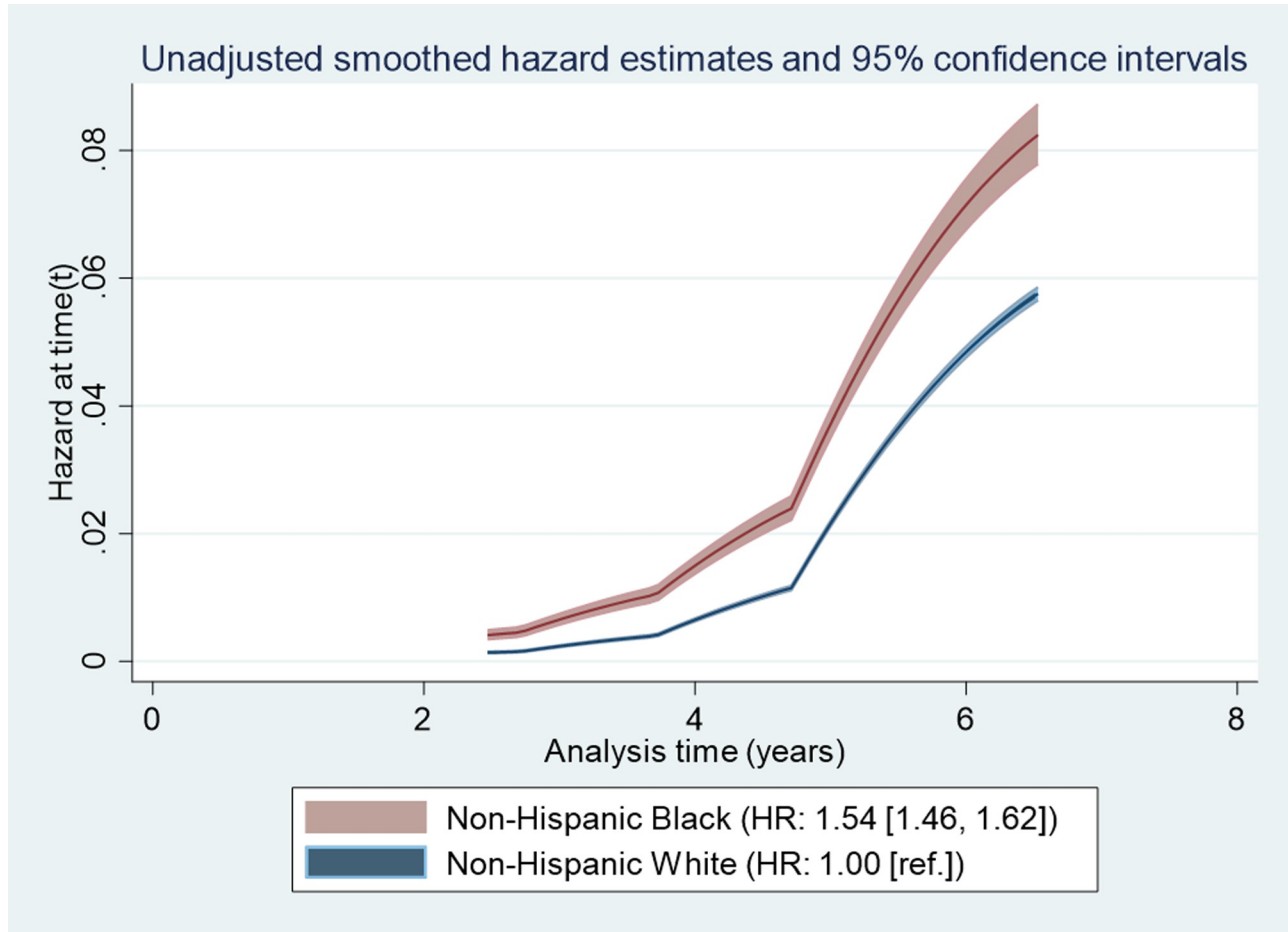

**Fig 1. Unadjusted smooth hazards and 95% confidence intervals (shaded) by race/ethnicity among non-Hispanic White and non-Hispanic Black respondents from the Women's Health Initiative Observational Study.**

**Table 2. Overall and common referent hazard ratios and 95% confidence intervals comparing the hazard for ≥10% weight gain by baseline weight status in non-Hispanic Blacks and non-hispanics (n = 70,750)[1-3].**

| | Hazard ratios (95% confidence interval) | | | | | |
| --- | --- | --- | --- | --- | --- | --- |
| | Normal weight | Overweight | Obese class I | Obese class II | Obese class III | *p*-trend |
| Crude models | | | | | | |
| Overall | 1.00 (ref.) | 1.03 (0.99, 1.07) | 1.05 (1.00, 1.11) | 1.17 (1.09, 1.26) | 1.00 (0.90, 1.11) | <0.001 |
| Non-Hispanic White | 1.00 (ref.) | 1.02 (0.99, 1.06) | 1.03 (0.98, 1.09) | | 0.95 (0.85, 1.05) | 0.071 |
| Non-Hispanic Black[4] | 1.77 (1.60, 1.96) | 1.49 (1.37, 1.63) | 1.47 (1.32, 1.64) | 1.72 (1.45, 2.03) | 1.43 (1.17, 1.74) | 0.063 |
| Non-Hispanic Black[5] | 1.00 (ref.) | 0.84 (0.74, 0.95) | 0.82 (0.71, 0.94) | 0.94 (0.78, 1.14) | 0.79 (0.63, 0.97) | 0.063 |
| Adjusted models | | | | | | |
| Overall | 1.00 (ref.) | 1.01 (0.97, 1.05) | 0.97 (0.92, 1.02) | 1.00 (0.93, 1.08) | 0.80 (0.72, 0.88) | 0.001 |
| Non-Hispanic White | 1.00 (ref.) | 1.01 (0.97, 1.05) | 0.97 (0.92, 1.02) | 0.99 (0.91, 1.06) | 0.77 (0.70, 0.86) | <0.001 |
| Non-Hispanic Black[4] | 1.45 (1.31, 1.60) | 1.21 (1.11, 1.33) | 1.15 (1.03, 1.28) | 1.26 (1.06, 1.49) | 1.02 (0.83, 1.24) | 0.005 |
| Non-Hispanic Black[5] | 1.00 (ref.) | 0.83 (0.74, 0.95) | 0.79 (0.69, 0.91) | 0.88 (0.73, 1.07) | 0.72 (0.58, 0.90) | 0.005 |
| Relative excess hazard, additive interaction[7] | -0.30 | -0.33 | -0.18 | -0.29 | | |
| Relative hazard due to multiplicative interaction[8] | 0.83 | 0.81 | 0.86 | 0.85 | | |

[1] Weight status was defined using baseline body mass index (BMI) as normal weight (BMI: 18.5–24.9 kg/m$^2$), overweight (BMI: 25.0–29.9 kg/m$^2$), obese class I (BMI: 30.0–34.9 kg/m$^2$), obese class II (BMI: 35.0–39.9 kg/m$^2$), or obese class III (BMI ≥40.0 kg/m$^2$).

[2] Adjusted models controlled for education level, annual household income, smoking status, alcohol intake, age, total energy intake at baseline and MET-hours of mild, moderate and hard exercise.

[3] *P*-trend corresponds to a Wald test statistic when a linear term for baseline body weight status was substituted in the model.

[4] Values shown are relative to the common referent group, normal weight non-Hispanic Whites.

[5] Values shown are relative to the referent group, normal weight non-Hispanic Blacks.

[6] Relative excess hazard due to additive interaction is based on adjusted models and defined as the hazard for weight gain in the doubly exposed less the sum of the null value (1) and the risk of weight gain in each singly exposed group. A value less than or greater than 0 would suggest departure from additive interaction.

[7] The relative hazard due to multiplicative interaction is based on adjusted models and defined as the ratio of the hazard in the doubly exposed to the product of hazards for each singly exposed group. A value less than or greater than 1 would suggest departure from multiplicative interaction.

were 1.17 (95% CI: 1.09, 1.26) times more likely to experience ≥10% weight gain than those who were normal weight at baseline. The overall trend for a linear term for baseline weight status was significant (p<0.001), but the directionality was not consistent across categories of baseline weight status. In adjusted models of the overall relationship between baseline weight status and risk of weight gain, HRs were significantly attenuated and the relationship between class II obesity and risk of weight gain was no longer significant. However, those with class III obesity at baseline were less likely (HR: 0.80; 95% CI: 0.72, 0.88) to than those who were normal weight at baseline. The was evidence of an inverse trend, which was significant (p = 0.001).

Table 2 also shows the results of common referent models comparing risk of weight gain in NHBs and NHWs to that of NHWs with normal weight at baseline. In unadjusted models, NHWs with class II obesity were more likely (HR: 1.13; 95% CI: 1.04, 1.22) to experience ≥10% weight gain. No other significant relationships were observed and the trend was not significant (p = 0.071). In adjusted models, the relationship between class II obesity and risk of weight gain in NHWs was no longer significant. The was evidence of an inverse trend, which was significant (p<0.001). However, NHWs with class III obesity at baseline were less likely (HR: 0.77; 95% CI: 0.70, 0.86) to experience ≥10% weight gain. In unadjusted models, NHBs who were normal weight (HR: 1.77; 95% CI: 1.60, 1.96), overweight (HR: 1.49; 95% CI: 1.37, 1.63), class I obesity (HR: 1.47; 95% CI: 1.32, 1.64), class II obesity (HR: 1.72; 95% CI: 1.45, 2.03) and class III obesity (HR: 1.43; 95% CI: 1.17, 1.74) were more likely to experience ≥10%

weight gain than normal weight NHBs. These relationships were attenuated in adjusted models. In unadjusted models, the trend was not significant in NHBs (p = 0.063). NHW with normal weight (HR: 1.45; 95% CI: 1.31, 1.60), overweight (HR: 1.21; 95% CI: 1.11, 1.33), class I obesity (HR: 1.15; 95% CI: 1.03, 1.28) and class II obesity (HR: 1.26; 95% CI: 1.06, 1.49) were more likely to experience ≥10% weight gain than NHWs with normal weight at baseline. The relationship between class III obesity and risk of weight gain in NHBs was no longer significant in adjusted models. There was evidence of an inverse trend, which was significant (p = 0.005).

Also presented in Table 2 are within strata HRs comparing risk of weight gain in NHBs to those who were normal weight at baseline. HRs in unadjusted and adjusted models were similar, but there was no significant trend in unadjusted models. In adjusted models, NHBs with overweight (HR: 0.83; 95% CI: 0.74; 0.95), class I obesity (HR: 0.79; 95% CI: 0.69, 0.91) and class III obesity (HR: 0.72; 95% CI: 0.58, 0.90) were less likely to experience ≥10% weight gain than NHBs who were normal weight at baseline. The trend was significant (p = 0.005), but the directionality of the trend was not clear. In both crude and adjusted models, there was no significant relationship observed for NHBs with class II obesity at baseline.

The relative excess risk due to additive interaction, and the proportion of risk due to multiplicative interaction are also presented in Table 2. As shown, the interaction of NHB race/ethnicity and weight status at baseline was less than additive and less than multiplicative.

Findings stratified by age group (49 to 54, 55 to 59, 60 to 64 and 65 and older) are presented in S1 Table. As in unstratified models, NHBs were more likely to experience ≥10% weight gain than NHWs in the same category of baseline weight status. Results were similar in unadjusted and adjusted models. Findings from age-stratified overall models and stratum-specific models comparing the risk of weight gain in NHBs to those who were normal weight at baseline are presented in S2 Table. Results of these models were also consistent with those of models not stratified by age.

## Discussion

NHB postmenopausal women were more likely to experience ≥10% weight gain than NHWs, which is consistent with findings from previous studies [10–13]. While this was true across all categories of baseline weight status, the difference in risk was most prominent in women with normal weight at baseline, with NHBs being more than 50% more likely to experience ≥10% weight gain than NHWs. This finding suggests that efforts to reducing the disparity in the prevalence of postmenopausal obesity among NHBs and NHWs should focus on preventing excess weight gain in NHBs with normal weight. Moreover, NHB postmenopausal women in our study had greater rates of class I, II and III obesity than NHWs, thereby suggesting that racial/ethnic divergence in the prevalence of obesity in NHBs and NHWs may have begun prior to study enrollment. A number of prior studies have also reported that in young-, middle-, and older- aged women, NHBs were at greater risk of weight gain than NHWs [10–13]. Moreover, NHB women enter middle age at a higher BMI, and gain less weight thereafter, than NHW women [12]. Accordingly, interventions to prevent weight gain at earlier ages would be instrumental to reducing racial/ethnic disparities in the prevalence of postmenopausal obesity.

Like others [26], we found that risk of weight gain was lower in women with obesity at baseline than those with normal weight. This was true in both NHBs and NHWs. With regard to the joint influence of NHB race/ethnicity and weight status at baseline on risk of postmenopausal weight gain, we found that overall, the relationship was both sub-additive and sub-multiplicative. Moreover, the overall greater risk of weight gain in NHBs vs. NHWs was due to a sharply higher risk of weight gain in normal weight NHBs. However, the risk of weight gain

was higher in NHBs across all categories of baseline weight status. This would suggest that the overall higher risk of weight gain in NHBs was not due to differences in weight status alone, but rather due to other biological [27, 28], social [10, 13, 29], or environmental factors [30, 31]. Biological differences may include differences in energy expenditure and fatty acid metabolism that promote excess weight gain in NHBs relative to NHWs [32–34]. Sociocultural differences between NHBs and NHWs may also contribute to the disparity in weight status. These include sociocultural differences in perceived weight status–it has been established that there are racial/ethnic differences in perceived weight status in NHB and NHW women. NHBs are more likely to perceive themselves as having a healthy body weight, even at higher BMIs, compared with NHWs [35], and this could contribute to higher rates of obesity among NHB women. Racial/ethnic differences in other sociocultural factors may also play a role. NHB women are more likely than white women to experience lower socioeconomic positioning, thereby increasing their likelihood of weight gain throughout the lifespan [11]. Other sociocultural differences in education level [36], annual household income [37], and physical activity level [38], have also been noted and may also contribute to the disparity in obesity prevalence among postmenopausal NHB and NHW women, along with environmental factors. NHBs are more likely to reside in neighborhoods with limited access to healthy foods [39], and limited neighborhood walkability [40], both of which are associated with increased risk of weight gain. Lastly, it has been shown that differences in weight gain throughout the life-cycle may also contribute to racial/ethnic disparity in obesity among postmenopausal women. NHB women experience greater weight gain than NHWs in early [41], middle [11, 42], and older adulthood [43], thereby placing them at greater risk of obesity than NHWs at virtually every age. This is consistent with the findings of our study. We found that postmenopausal NHBs had markedly higher rates of class I obesity, and more than twice the rates of class II and class III obesity, as NHWs. Recent evidence from the US nationally representative National Health and Nutrition Examination Survey (NHANES), which showed that in middle-(40–59 y) and older-(60+ y) aged women, the prevalence of obesity was an average of 20 percentage points higher in NHBs compared to NHWs (1), also supports this result. Taken together, these findings suggest that much of the divergence in the prevalence of obesity in NHW and NHB postmenopausal women occurs prior to age 40. Accordingly, efforts to reduce racial/ethnic disparities in obesity will require a focus on preventing excess weight gain in NHB women at earlier life stages, and particularly in those younger than age 40.

There are a number of important limitations to our study that bear mentioning here. Foremost, the study sample may not have been representative of the current population of US postmenopausal women. Notably, women in the WHI OS were ages 49 to 81 years upon enrollment, which completed in 1998, and study participants were recruited from 40 clinical centers across 24 US states and the District of Columbia, which may have limited the representativeness of the sample. A further limitation is the potential for misreporting of dietary intake due by measuring dietary intake using an FFQ, which may be more prone to misreporting than other self-report measures such as a 24-hour recall or food records [44]. Common sources of error include the comprehensiveness of the food list and the length of time over which respondents are asked to recall their diet, and the total time it takes to complete the questionnaire [45, 46]. Nonetheless the WHI FFQ has been validated in a number of studies, which have shown the WHI FFQ to have acceptable correlations with other common measures of diet [47]. Moreover, as the current study was interested in capturing 'usual' dietary intake, the FFQ is more aptly suited than measures designed to measure diet over short periods.

To characterize the primary outcome self-reported weight was used, which may be prone to both intentional and unintentional misreporting [48, 49]. Moreover, weight was measured only at baseline, while self-reported 'highest weight since last follow-up' was ascertained at all

subsequent years of follow-up starting with year one. Accordingly, the self-report measure was used in conjunction with measured weight (at baseline) to compute percent change in weight from baseline. Nonetheless, we found measured weight at baseline to have acceptable correlation with highest reported weight in the time since last follow-up at year one. Finally, it should be noted that the average age of enrollment into the WHI OS was 63.7 years, whereas it has been previously reported that the median age of menopause among US women is 52.6 years [50]. Accordingly, for some women in this study, the period immediately after the onset of menopause may not have been captured, which limits our ability to draw conclusions about weight gain during the period shortly after menopause. Lastly, it is a limitation of this study that the outcome was defined as a relative measure ($\geq$10%). This definition of significant weight gain corresponds to the widely-used definition of significant weight loss ($\geq$10%) as defined by Wing and Hill of the National Weight Loss Consortium [51]. Use of a relative measure of weight loss meant that women who were heavier at baseline would need to gain more weight than those who weighed less in order to achieve a 10% weight gain. Nonetheless, in sensitivity analysis using $\geq$10 pound weight gain as the primary outcome, the results were similar to those presented from the primary analysis (see S4 Table).

## Conclusion

This study provides several important contributions to the literature. First, there are few studies that explore differences in the risk of postmenopausal weight gain in NHBs and NHWs. Consistent with prior studies we found that, overall, NHB women were more likely to experience $\geq$10% weight gain than NHW women. This was true in every category of baseline weight status, and the difference in risk was especially pronounced in women with normal weight at baseline. This finding suggests that efforts to prevent postmenopausal weight gain in NHBs would be best directed toward those who are normal weight. Furthermore, we found that NHB postmenopausal women in our sample had significantly higher rates of class I, II, and III obesity than NHWs at baseline, suggesting that divergence in the prevalence of obesity begins prior to menopause. Future studies seeking to address the disparity in postmenopausal obesity should focus on preventing weight gain in NHBs prior to menopause. Reducing the rates of obesity in NHB postmenopausal women could help to reduce racial/ethnic disparities in risk of obesity-related chronic diseases in NHBs and NHWs.

## Supporting information

**S1 Fig. Unadjusted smooth hazards by weight status among non-Hispanic White and non-Hispanic Black respondents from the Women's Health Initiative Observational Study.** (TIF)

**S1 Table. Common referent hazard ratios and 95% confidence intervals comparing the hazard of a $\geq$10% weight gain by baseline weight status in non-Hispanic Blacks and non-Hispanic Whites stratified by age (n = 70,750)** [1–3]. (DOCX)

**S2 Table. Overall and stratum-specific hazard ratios and 95% confidence intervals comparing the hazard for $\geq$10% weight gain by baseline weight status in non-Hispanic Blacks and non-hispanics stratified by age (n = 70,750)** [1-3]. (DOCX)

**S3 Table. Overall and common referent hazard ratios and 95% confidence intervals comparing the hazard for $\geq$10% weight gain by baseline weight status in non-Hispanic Blacks**

and non-hispanics using complete-case analysis [1–3].
(DOCX)

**S4 Table. Overall and common referent hazard ratios and 95% confidence intervals comparing the hazard for ≥10 pound weight gain by baseline weight status in non-Hispanic Blacks and non-hispanics [1–3].**
(DOCX)

## Author Contributions

**Conceptualization:** Christopher N. Ford, Alexis C. Wood.

**Data curation:** Christopher N. Ford, JoAnn E. Manson.

**Formal analysis:** Christopher N. Ford.

**Funding acquisition:** Chloe E. Bird, Mara Z. Vitolins.

**Investigation:** Christopher N. Ford, Alexis C. Wood, Chloe E. Bird, Mara Z. Vitolins.

**Project administration:** Christopher N. Ford, Shine Chang, Alexis C. Wood, Mara Z. Vitolins.

**Supervision:** Shine Chang, Alexis C. Wood, JoAnn E. Manson, Helena Laroche.

**Writing – original draft:** Christopher N. Ford.

**Writing – review & editing:** Christopher N. Ford, Shine Chang, Alexis C. Wood, JoAnn E. Manson, David O. Garcia, Helena Laroche, Chloe E. Bird, Mara Z. Vitolins.

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
