## [Decision Letter · Decision Letter 0]

14 Oct 2020

PONE-D-20-26567

On the joint role of non-Hispanic Black race/ethnicity and weight status in predicting

postmenopausal weight gain

PLOS ONE

Dear Dr. Ford,

Thank you for submitting your manuscript to PLOS ONE. After careful consideration, we feel that it has merit but does not fully meet PLOS ONE’s publication criteria as it currently stands. Therefore, we invite you to submit a revised version of the manuscript that addresses the points raised during the review process.

While the reviewers found the paper interesting, they have raised a number of content and methodological issues. Each reviewer has provided detailed feedback about these issues.

We look forward to receiving your revised manuscript.

Kind regards,

Luisa N. Borrell, DDS, PhD

Academic Editor

PLOS ONE

Journal Requirements:

"Study procedures were approved by institutional review boards at each participating Women's Health Initiative study site."

For additional information about PLOS ONE ethical requirements for human subjects research, please refer to " ext-link-type="uri" xlink:type="simple">http://journals.plos.org/plosone/s/submission-guidelines#loc-human-subjects-research."

Reviewers' comments:

Reviewer's Responses to Questions

**Comments to the Author**

1. Is the manuscript technically sound, and do the data support the conclusions?

Reviewer #1: Yes

Reviewer #2: Partly

2. Has the statistical analysis been performed appropriately and rigorously? 

Reviewer #1: Yes

Reviewer #2: N/A

3. Have the authors made all data underlying the findings in their manuscript fully available?

Reviewer #1: Yes

Reviewer #2: Yes

4. Is the manuscript presented in an intelligible fashion and written in standard English?

Reviewer #1: Yes

Reviewer #2: Yes

5. Review Comments to the Author

Reviewer #1: The authors have been responsive to the reviewers' comments and have appropriately incorporated their suggestions, but there are still a few concerns that need to be addressed.

1. Page 8, Lines 197 -199. The result for normal weight NHB, which was the strongest association of all the weight categories, was not described in the text. Also, the HR reported for class III obesity should be 1.39 (1.14,1.71) not 0.74 (0.59,0.92).

2. In this same paragraph, lines 192 to 205, the authors should mention the direction of the trends when they report the p-values. It appears to be a direct trend for the overall group and the NHW, but an inverse trend for the NHB.

3. There are some numbers in the table that seem to have been included in error -- 01 after the HR and CI for NHB normal weight, 10 after the HR and CI for NHW overweight and 11 after the HR and CI for NHB overweight.

4. Page 9, paragraph starting with line 225. I'm confused about this paragraph. These data are supposed to be in Table 2, but I don't see any of these numbers in this table.

5. Minor comment, Page 7, line 183. The sentence begins with "for the" rather than "The".

Reviewer #2: 1) I wonder if any respondents with disease, such as cancer, were also excluded from the sample for analysis. This is a serious condition that might affect weight gain/loss over time.

2) It is reported that 965 cases are missing in BMI so excluded from the sample. However, there are a couple of statistical methods that can handle missing data, such as multiple imputations (MI) and full information maximum likelihood (FIML). In my opinion, it is necessary to compare the results with and without missing data and report the one without missing data.

3) No information was provided for missing data of the variables in Table 1. This is crucial since it would affect sample size for each model in Table 2 and Appendix Table 1. Some variables seem to have missing data, such as household income. For example, numbers for household income of NHW does not add up to 76,208 (which is not equal to 70,909=9,622+31,183+21,744+8,360). Please report the total sample size for each variable in Table 1. Ideally, this can be handled properly by MI or FIML.

4) Please report the sample size for each analysis in Table 2 and Appendix Table 1.

5) Since additional analysis (Appendix Table 1) considered both race/ethnicity and age groups, please provide the sample sizes for age groups in Table 1. Since the average ages reported in Table 1 are over 62, this might result in very small sample size for some categories, particularly for younger age groups for NHB, which might lead to low statistical power for some categories. This might be the reason of several nonsignificant results for NHB in Appendix Table 1.

6) Two exercise variables in the final adjusted model are likely to be highly correlated with each other. For that reason, it would be better if a global scale can be controlled for, instead of individual scales. Same concern over caloric intake at baseline and year 3 follow-up.

7) On page 7, it says that final adjusted models include several variables. However, in Table 1, there are additional variables not mentioned, such as waist circumference and education. This might confuse readers. If they were not used in the final models, it would be better not to be included in Table 1 for descriptive statistics.

8) Waist circumference can also be used to assess obesity. In fact, it might be more meaningful than BMI: https://www.hsph.harvard.edu/obesity-prevention-source/obesity-definition/abdominal-obesity/. If it was measured more than one time, like weight and height, then it would be extremely meaningful if both BMI and waist circumference can be analyzed and compared across race/ethnicity.

9) Menopause, on average, is likely to happen at early 50s so linking menopause with weight gain among women in this data seems to be a very ambitious stretch. Because of this reason, it is crucial to carefully examine the patterns of age 50-54 for both NHW and NHB, but there is a concern over sample size particularly for this age group of NHB. As a matter of fact, there are few significant results once adjusted for several factors especially when NHB was compared with NHW in the same weight status. I think the introduction needs to be revised so that it can actually lead to the main hypotheses for this study.

10) In Appendix Table 1, readers need to compare the numbers horizontally and vertically for NHB but it is very confusing. Please consider summarizing numbers in a separate table for the comparison between NHB and NHW or reorganize Appendix Table 1 for better readability.

11) Please do not use a subjective term, such as “just” on page 3 line 75.

12) In table 2, it seems that there are some numbers that might not belong to the table. Check the numbers positioned next to 95% C.I. for NHW and NHB under the adjusted models. Please review the numbers in tables.

6. PLOS authors have the option to publish the peer review history of their article (what does this mean?). If published, this will include your full peer review and any attached files.

Reviewer #1: No

Reviewer #2: No

---

## [Author Response · Author response to Decision Letter 0]

30 Dec 2020

Please see our attached Response to Reviewers.

---

## [Decision Letter · Decision Letter 1]

25 Jan 2021

PONE-D-20-26567R1

On the joint role of non-Hispanic Black race/ethnicity and weight status in predicting

postmenopausal weight gain

PLOS ONE

Dear Dr. Ford,

Thank you for submitting your manuscript to PLOS ONE. After careful consideration, we feel that it has merit but does not fully meet PLOS ONE’s publication criteria as it currently stands. Therefore, we invite you to submit a revised version of the manuscript that addresses the points raised during the review process.

ACADEMIC EDITOR:

While the paper shows some improvements, the reviewers have found some old and new issues with the paper. I will strongly encourage you to pay careful attention to the specific comments provided by each reviewer below.

We look forward to receiving your revised manuscript.

Kind regards,

Luisa N. Borrell, DDS, PhD

Academic Editor

PLOS ONE

Reviewers' comments:

Reviewer's Responses to Questions

**Comments to the Author**

1. If the authors have adequately addressed your comments raised in a previous round of review and you feel that this manuscript is now acceptable for publication, you may indicate that here to bypass the “Comments to the Author” section, enter your conflict of interest statement in the “Confidential to Editor” section, and submit your "Accept" recommendation.

Reviewer #1: (No Response)

Reviewer #2: All comments have been addressed

2. Is the manuscript technically sound, and do the data support the conclusions?

Reviewer #1: Partly

Reviewer #2: Yes

3. Has the statistical analysis been performed appropriately and rigorously? 

Reviewer #1: I Don't Know

Reviewer #2: Yes

4. Have the authors made all data underlying the findings in their manuscript fully available?

Reviewer #1: Yes

Reviewer #2: Yes

5. Is the manuscript presented in an intelligible fashion and written in standard English?

Reviewer #1: Yes

Reviewer #2: Yes

6. Review Comments to the Author

Reviewer #1: In re-reviewing this manuscript, I realized that the definition of the primary outcome raised a significant concern in the interpretation of the results, specifically the results stratified by baseline weight. The outcome is defined as 10% increase in weight from baseline, which means that, for example, a woman who started out at a weight of 120 lbs (normal BMI), would meet the threshold for the outcome if she gained 13 pounds, whereas a woman who had a baseline weight of 250 lbs (obese class III) would have to gain at least 25 pounds. To conclude that “the risk of weight gain was lower in women with obesity at baseline than those with normal weight” may not be accurate. It is possible that the very obese women gained on average 20 lbs but many of them did not meet the 10% increase threshold while the normal weight women could have had an average weight gain of 15 lbs, with more of them meeting the 10% threshold.

At a minimum, the authors should revise the manuscript to describe the outcomes more precisely, i.e., in the text they should consistently describe the outcome as “risk of 10% weight gain” rather than simply “risk of weight gain”. The discussion should also acknowledge the limitations of defining the outcome as a 10% increase in weight rather than an absolute change in weight.

Preferably, the authors should re-analyze the data with the outcome as a continuous variable, looking at the absolute change in weight and determining if the conclusions are similar to those reached when defining the outcome as a weight gain of at least 10% change from baseline.

In addition to this major concern, I have the following comments:

1. Introduction, lines 24-27: This is a run-on sentence and its meaning is not entirely clear.

2. Approach, line 75: Delete the word “in”.

3. Results, line 129: As described above, the results should be described more precisely “…were more likely to have a weight gain of 10%” here and throughout the paper.

4. Results, line 141-162: Many of the numbers in these two paragraphs do not exactly match the numbers in Table 2. All numbers should be checked for accuracy.

5. Results, line 151: NHW in this line should be NHB.

6. Results, line 157: It is not clear what is meant when the authors state “HRs in unadjusted models were similar..” Similar to what?

Reviewer #2: 1) Please provide the reason for the choice of 10 data sets, not 20 or 5, for multiple imputations briefly in text. Also, please report what statistical program was used for analysis.

2) Please briefly discuss the differences in the results with and without multiple imputations. It is important to know if there are substantially different estimations between two approaches (listwise vs. MI).

3) In Table 2 and Appendix Table 1, the reference group is always NHW with normal weight. Please provide the reasoning for the choice. Why not compare NHB with NHW within each weight status group? It would be more interesting, for example, if NHB with overweight was compared to NHB with overweight, not to NHW with normal weight.

4) Please provide the reasoning for 10% or more increase in weight as the outcome. The study is assuming that 10% or more weight gain for someone with normal weight at baseline has the same effect for someone with overweight or obesity at baseline. But gaining 10% or more weight would be much more difficult and harmful for someone with obesity than for someone with normal weight.

7. PLOS authors have the option to publish the peer review history of their article (what does this mean?). If published, this will include your full peer review and any attached files.

Reviewer #1: No

Reviewer #2: No

---

## [Author Response · Author response to Decision Letter 1]

28 Jan 2021

Please see our Response to Reviewer Comments.

---

## [Decision Letter · Decision Letter 2]

16 Feb 2021

On the joint role of non-Hispanic Black race/ethnicity and weight status in predicting

postmenopausal weight gain

PONE-D-20-26567R2

Dear Dr. Ford,

We’re pleased to inform you that your manuscript has been judged scientifically suitable for publication and will be formally accepted for publication once it meets all outstanding technical requirements.

Kind regards,

Luisa N. Borrell, DDS, PhD

Academic Editor

PLOS ONE

Reviewers' comments:

Reviewer's Responses to Questions

**Comments to the Author**

1. If the authors have adequately addressed your comments raised in a previous round of review and you feel that this manuscript is now acceptable for publication, you may indicate that here to bypass the “Comments to the Author” section, enter your conflict of interest statement in the “Confidential to Editor” section, and submit your "Accept" recommendation.

Reviewer #1: All comments have been addressed

Reviewer #2: All comments have been addressed

2. Is the manuscript technically sound, and do the data support the conclusions?

Reviewer #1: (No Response)

Reviewer #2: Yes

3. Has the statistical analysis been performed appropriately and rigorously? 

Reviewer #1: (No Response)

Reviewer #2: Yes

4. Have the authors made all data underlying the findings in their manuscript fully available?

Reviewer #1: (No Response)

Reviewer #2: Yes

5. Is the manuscript presented in an intelligible fashion and written in standard English?

Reviewer #1: (No Response)

Reviewer #2: Yes

6. Review Comments to the Author

Reviewer #1: The authors have adequately addressed the concerns raised in the last review. No additional comments.

Reviewer #2: I really appreciate the authors for taking time to rigorously address all my comments throughout multiple revisions. It is a very interesting and meaningful study, and I am sure that many scholars would find it helpful.

7. PLOS authors have the option to publish the peer review history of their article (what does this mean?). If published, this will include your full peer review and any attached files.

Reviewer #1: No

Reviewer #2: No

---

## [Editor Report · Acceptance letter]

17 Feb 2021

PONE-D-20-26567R2 

On the joint role of non-Hispanic Black race/ethnicity and weight status in predicting postmenopausal weight gain 

Dear Dr. Ford:

I'm pleased to inform you that your manuscript has been deemed suitable for publication in PLOS ONE. Congratulations! Your manuscript is now with our production department. 

Kind regards, 

on behalf of

Dr. Luisa N. Borrell 

Academic Editor

PLOS ONE